# The Proton Leak of the Inner Mitochondrial Membrane Is Enlarged in Freshly Isolated Pancreatic Islets

**DOI:** 10.3390/biomedicines12081747

**Published:** 2024-08-02

**Authors:** Mohammed Alshafei, Mai Morsi, Julia Reschke, Ingo Rustenbeck

**Affiliations:** 1Institute of Pharmacology, Toxicology and Clinical Pharmacy, Technische Universität Braunschweig, 38106 Braunschweig, Germany; mohammed.alshafei@tu-bs.de (M.A.); mai.morsi@yahoo.com (M.M.); julia.reschke@tu-bs.de (J.R.); 2Department of Pharmacology, Faculty of Pharmacy, Assiut University, Assiut 71526, Egypt; 3PVZ-Center of Pharmaceutical Engineering, Technische Universität Braunschweig, 38106 Braunschweig, Germany

**Keywords:** insulin secretion, metabolic amplification, mitochondrial membrane potential, oxygen consumption, pancreatic islets

## Abstract

In a number of investigations on the mechanism of the metabolic amplification of insulin secretion, differences between the response of freshly isolated islets and of islets cultured for one day have been observed. Since no trivial explanation like insufficient numbers of viable cells after cell culture could be found, a more thorough investigation into the mechanisms responsible for the difference was made, concentrating on the function of the mitochondria as the site where the metabolism of nutrient stimulators of secretion forms the signals impacting on the transport and fusion of insulin granules. Using combinations of inhibitors of oxidative phosphorylation, we come to the conclusion that the mitochondrial membrane potential is lower and the exchange of mitochondrial reducing equivalents is faster in freshly isolated islets than in cultured islets. The significantly higher rate of oxygen consumption in fresh islets than in cultured islets (13 vs. 8 pmol/min/islet) was not caused by a different activity of the F_1_F_0_-ATPase, but by a larger proton leak. These observations raise the questions as to whether the proton leak is a physiologically regulated pathway and whether its larger size in fresh islets reflects the working condition of the islets within the pancreas.

## 1. Introduction

The pancreatic beta cell senses the availability of nutrients and transforms this signal into insulin secretion rates which need to be adequate for the requirement of the intact organism. To this end, a modulation of the secretory response by hormones and the autonomic nervous system is necessary. But it is the elevated glucose concentration which under physiological conditions is the condition for the increase in insulin secretion. This role of glucose can be mimicked by a few carbohydrates and a limited number of amino- and keto acids, which makes them interesting as experimental tools, but is not of physiological relevance because of their low concentrations in the mammalian organism [1,2].

A consensus exists that the mitochondrial metabolism of the beta cell is the site where the metabolic pathways of the insulinotropic nutrients converge [3] (see however [4]) and where the intracellular signals regulating the secretion of insulin are generated. It is generally recognized that the increase in the cytosolic ATP/ADP ratio by increased oxidative phosphorylation induces the electrical activity of the beta cell by closure of the K_ATP_ channels [5,6]. Forcing these channels in an open position by pharmacological agents effectively suppresses nutrient-induced insulin secretion [7,8].

While the action potential spiking and its underlying event, the Ca^2+^ influx via voltage-dependent Ca^2+^ channels, are necessary for the onset of the stimulated secretory activity [9,10], another pathway, in addition to this triggering signal, is apparently operative to amplify the extent of secretion. The metabolic amplification of insulin secretion is the amount of insulin released by beta cells in response to nutrient secretagogues which surpasses the amount released in response to purely depolarizing stimuli, such as K_ATP_ channel-closing agents or the increase in the extracellular K^+^ concentration [11]. What is generally accepted is that the mitochondrial metabolism of the nutrients is required for the metabolic amplification. [12]. In particular, the cataplerotic export of mitochondrial metabolites seems to be involved in this pathway [13,14].

While investigating the mechanisms involved in the metabolic amplification we realized that the secretory response of freshly isolated mouse islets to nutrient secretagogues, like glucose or alpha-ketoisocaproic acid (KIC), differed from the one of cultured islets in a number of respects. Specifically, the following observations in preceding investigations prompted us to investigate the mitochondrial membrane potential of fresh and cultured islets in more detail: 1. Fresh islets respond with a moderate first phase followed by an ascending second phase to the square wave increase in glucose from 0 to 30 mM, whereas cultured islets respond to this stimulation with a marked first phase response followed by an elevated plateau [15]. 2. At basal and at stimulatory glucose concentrations, fresh islets had a significantly higher oxygen consumption rate (OCR) than cultured islets [16]. 3. The inhibition of cytosolic protein translation by cycloheximide in fresh islets resulted in a marked inhibition of glucose-induced insulin secretion, but did not inhibit the secretory response of cultured islets [16].

By using inhibitors of oxidative phosphorylation with different mechanisms of action we sought to identify the different working conditions of beta cell mitochondria in fresh and in cultured islets. Taken together, the present observations suggest that mitochondria in freshly isolated islets have a higher leakiness of the inner membrane and in consequence a lower degree of energy coupling than mitochondria in cultured islets. This may be responsible, at least in part, for the different kinetics of insulin secretion.

## 2. Materials and Methods

*Chemicals.* Tetramethylrhodamine ethyl ester (TMRE) was from AnaSpec (Fremont, CA, USA). Carbonylcyanide–m-chlorphenylhydrazone (CCCP), cycloheximide, oligomycin, sodium azide (NaN_3_), collagenase P and the cell culture medium RPMI 1640 (without glucose) were from Sigma-Aldrich (Taufkirchen, Germany). Fetal calf serum (FCS Gold ADD) was obtained from Bio & Sell (Nürnberg-Feucht, Germany), bovine serum albumin (BSA, fraction V) and all other reagents of analytical grade were from E. Merck (Darmstadt, Germany).

*Tissue and tissue culture.* NMRI mouse islets were isolated from the pancreas of female mice (12–14 weeks) which had unrestricted access to phytoestrogen-free standard chow and water. After death by cervical dislocation, the collagenase solution was injected into the bile duct, thereafter, the entire pancreas was removed and digested in a shaking water bath. The released islets were collected under a stereomicroscope. The time between onset of the digestion and the start of the perifusion of freshly isolated was ca. 45 min. To enable a close comparison between fresh and cultured islets the same batch of islets was split in two one half for immediate use, the other half for use after culture. The culture medium was RPMI 1640 with 10% FCS and 5 mM glucose, the culture took place in a humidified atmosphere of 95% air and 5% CO_2_ at 37 °C, the culture duration was 22 ± 1 h. Animal care took place in the central animal facility of the Technische Universität Braunschweig and was supervised to conform to the current EU regulations by the state animal welfare authority (LAVES, Oldenburg, Germany). Without prior specific treatment, the sacrifice of rodents for ex vivo experimentation is not classified as animal experimentation, thus no project-specific registration number exists.

*Measurement of the NAD(P)H- and FAD-autofluorescence and of the TMRE-fluorescence.* During one perifusion experiment the fluorescence of NAD(P)H (=NADH + NADPH), of FAD [17,18], and of the indicator of the mitochondrial membrane potential, TMRE [19,20] was recorded. After loading with TMRE (20 nM for 40 min at 36 °C), one islet was placed in a custom-made perifusion chamber on the stage of an epifluorescence microscope (Leitz/Leica, Wetzlar, Germany) and perifused at 0.2 mL/min with Krebs–Ringer medium (KR medium), which was saturated with 95% O_2_ and 5% CO_2_ and contained (mM): NaCl 118.5, KCl 4.7, CaCl_2_ 2.5, KH_2_PO_4_ 1.2, MgSO_4_ 1.2, NaHCO_3_ 20, HEPES 10, BSA 0.2% *w*/*v*. The fluorescence was excited with a 150 W xenon arc using filter cubes (Omega Optical, Brattleboro, VT, USA) with the following characteristics: for NAD(P)H, excitation 366 ± 15 nm bandpass, dichroic separation 405 nm, emission 450 ± 32 nm bandpass, for FAD, excitation 440 ± 21 nm bandpass, dichroic separation 455 nm, emission 520 ± 20 nm bandpass, for TMRE, excitation 535 ± 18 nm bandpass, dichroic separation 570 nm, emission 590 ± 18 nm bandpass. The filter cubes were switched every 2.5 s with an exposure time of 0.1 s. A subregion of the perifused islet was selected for the fluorescence excitation, the emission was collected by a Zeiss Fluar objective (40x, 1.3 N.A.) and measured by a photon-counting multiplier. As a semiquantitative parameter of the level of mitochondrial reducing equivalents the NAD(P)H/FAD ratio was calculated [21]. For the comparison of single islet cells and intact islets the single cells were loaded with 20 nM TMRE for 20 min at 32 °C and the fluorescence of cells and islets was registered by a cooled CCD camera attached to a Zeiss Axiovert microscope. The mean values were calculated by normalizing the fluorescence intensity to 100% at the last prestimulatory time point in each single experiment. For the comparison of cycloheximide-exposed and -unexposed islets, these traces were multiplied with the mean intensity values at this time point.

*Measurement of islet oxygen consumption.* Miniaturized oxygen sensors (Pst3 sensors and Fibox4 m, PreSens, Regensburg, Germany) were placed at the inflow and the outflow of the islet perifusion chamber. The KR perifusion medium was constantly equilibrated with a gas mixture containing 21 vol% oxygen and passed an “artificial lung” equilibrator [22]. This resulted in constant oxygen levels which were registered by the first sensor at the chamber inflow. After perifusing the islets, the medium passed the second sensor. The difference between the sensors divided by the number of islets times the flow rate gave the oxygen consumption rate (OCR). To achieve stable recordings, a higher number of islets (150) and a slower pump rate (125 µL/min) were necessary than used for the secretion measurements [23]. The slower pump rate generated a lag time between the upstream and the downstream sensor, which had to be corrected for the calculation of the OCR kinetics.

*Statistics.* Results are presented as the mean ± SEM. GraphPad Prism5 software (GraphPad, LaJolla, CA, USA) was used for statistic calculations. If not stated otherwise “*t*-test” refers to the unpaired, two-sided *t*-test and “significant” refers to *p* < 0.05. A marginal significance (0.05 ≤ *p* ≤ 0.07) is indicated by a symbol in parentheses.

## 3. Results

NaN_3_ is a known as a fast-acting and reversible inhibitor of the oxidative phosphorylation (see, e.g., [24]). This property permitted an experimental protocol by which the islets were stimulated for 10 min by 25 mM glucose prior to the exposure to NaN_3_ and by repeating the glucose stimulation after wash-out of NaN_3_. 5 mM NaN_3_ caused an increase in the NAD(P)H-autofluorescence and a decrease in the NAD-autofluorescence which were nearly of the same extent and kinetics as the effect caused by the preceding stimulation by 25 mM glucose (Figure 1A). In contrast to the glucose stimulation, which caused an increase in the TMRE fluorescence, NaN_3_ caused a marked decrease. Like the effect on the autofluorescence, the decrease in the membrane potential was reversible upon wash-out. The repetition of the glucose stimulation led to practically the same effects as the first stimulation (Figure 1A).

When this protocol was performed with cultured islets, the effect size of the glucose stimulation was comparable to the one observed with fresh islets. Also, the observation of a virtually complete reversibility of the inhibitory effect of NaN_3_ could be confirmed (Figure 1B). However, the NaN_3_-induced changes in either type of autofluorescence were much less marked than those induced by glucose and the TMRE decrease by NaN_3_ was significantly smaller than the one in fresh islets (Figure 1B,C). The NaN_3_-induced changes in the TMRE fluorescence occurred at the same time as the changes in the autofluorescence while the TMRE response to glucose stimulation was retarded by approximately 5 min.

The difference between fresh and cultured islets was further examined by using the protonophore CCCP to inhibit the oxidative phosphorylation. In these experiments, the preceding perifusion with 25 mM glucose was prolonged to 20 min. This measure revealed a sudden deceleration of the autofluorescence response with cultured islets after ca. 10 min, whereas the autofluorescence of fresh islets changed steadily, generating an exponential saturation function (Figure 2A,B). Thus, fresh islets showed a significantly higher value at the end of the glucose stimulation, but also a faster decrease upon wash-out (Figure 2C).

With both types of islets, the addition of CCCP resulted in a fast decrease in NAD(P)H and TMRE, followed by a slow decrease which persisted during wash-out. The amplitude of the fast TMRE decrease was significantly larger in cultured islets than in fresh islets. In the subsequent period, a dissociation between the changes in the NAD(P)H and the FAD fluorescence was noted, in that the slow decrease and partial recovery of the NAD(P)H fluorescence was not mirrored by the FAD fluorescence, which remained unchanged.

These observations suggested that cells within fresh islets have a lower mitochondrial membrane potential than cells within cultured islets. This was tested by perifusion of the islets with CCCP in the presence of oligomycin. Prior to the addition of CCCP, the TMRE signal was in a steady state, whereas the NAD(P)H/FAD ratio as a measure of the reducing equivalents was slowly increasing (Figure 3A–D).

The difference between the fluorescence immediately before the addition of CCCP and the fluorescence after 20 min of perifusion with CCCP was considered as a measure of the mitochondrial membrane potential and of the level of the mitochondrial reducing equivalents. The reduction in the TMRE fluorescence by CCCP was always paralleled by reduced levels of the reducing equivalents. Both in the absence (Figure 3A,B) and the presence (Figure 3C,D) of the inhibitor of cytosolic translation, cycloheximide, the CCCP-induced reduction in TMRE fluorescence was significantly stronger in cultured islets than in fresh islets (Figure 4). The presence of cycloheximide slightly enhanced the decrease in the mitochondrial membrane potential and markedly enhanced the loss of the reducing equivalents (Figure 4).

When the glucose concentration in the perifusion medium was raised from a low value (1 mM), the TMRE fluorescence increase in cultured islets became significantly larger than the one of fresh islets within 10 min of stimulation (Figure 5A). The relative fluorescence decrease caused by CCCP in the presence of oligomycin was even larger in 1 day-cultured single islet cells than in islets cultured for the same time (Figure 5B). Both, a higher level prior to the addition of CCCP and a lower level after 20 min of CCCP perifusion contributed to this result. These observations suggest that (i) the non-mitochondrial binding of TMRE is larger in islets and likely contributes to the different minimal values in the experiments depicted in Figure 3 and that (ii) the difference in the mitochondrial membrane potential is not only generated by mitochondrial inhibitors but also reflects alterations in glucose metabolism.

The mechanism underlying the different membrane potential of fresh and cultured islets was investigated by stepwise blocking the respiratory chain while measuring the oxygen consumption rate (OCR, Figure 6 and Figure 7). To enable marked reductions in the OCR measurements, 25 mM glucose was used in these experiments. Throughout the perifusion, the OCR of the fresh islets was significantly higher than that by cultured islets (Figure 6A,B).

The OCR decrease by oligomycin was not significantly different between fresh and cultured islets, but the OCR decrease by antimycin plus rotenone was larger in fresh than in cultured islets (Figure 6C). This experiment was repeated in the presence of cycloheximide (Figure 7) and the difference was even more marked. The OCR of the freshly isolated islets was nearly twice as large as that by cultured islets (Figure 7A,B). Again, the OCR decrease in response to oligomycin was not significantly different between fresh and cultured islets, but the OCR decrease by antimycin plus rotenone was markedly larger in fresh than in cultured islets (Figure 7C).

## 4. Discussion

The present observations suggest that freshly isolated islets have a lower mitochondrial membrane potential and a higher OCR than islets cultured for one day because of a larger leak current across the inner mitochondrial membrane. Using inhibitors of specific sites within the electron transport chain, the steps controlling mitochondrial membrane potential and OCR were studied. The relevance of this observation may not be restricted to the technique of islet isolation but may point to more fundamental issues of beta cell function.

The inhibition of the respiratory chain at multiple sites by NaN_3_, predominantly at the cytochrome oxidase [25,26,27] leads to a backlog of reducing equivalents which explains the increase in the NAD(P)H and the decrease in the FAD fluorescence (see Figure 1). The decrease in the mitochondrial membrane potential (TMRE fluorescence in Figure 1) is not a direct action of NaN_3_, but reveals the leakiness of the inner mitochondrial membrane for protons when the proton-pumping activity of the respiratory chain is blocked. This explains why the NaN_3_-induced TMRE decrease is slower than the one produced by protonophoric uncouplers like CCCP (compare Figure 1 and Figure 2).

Since it can be assumed that the action of NaN_3_ is of the same magnitude in fresh and in cultured islets, the much smaller NaN_3_-induced changes in the autofluorescence in cultured than in fresh islets likely results from a lower production rate of reducing equivalents in the cultured islets. The similar level of reducing equivalents reached during glucose stimulation of fresh and cultured islets appears to contradict this explanation; however, it can be reconciled by the balance between lowered production by the Krebs cycle and lower usage by the respiratory chain of cultured islets. The lower consumption of reducing equivalents fits to the observation of a lower OCR of cultured islets (see Figure 6 and Figure 7).

The main observation generated using the protonophore CCCP is the larger extent of the fast TMRE fluorescence decrease in cultured islets. This may seem to be in conflict with the smaller TMRE fluorescence decrease as generated by NaN_3_, but the protonophore-mediated decrease reflects the mitochondrial membrane potential whereas the NaN_3_-induced decrease reflects the intrinsic leakiness. Thus, the use of these inhibitors permits the conclusion that the mitochondria in cultured islets are less leaky and have a higher membrane potential. The prolonged perifusion with 25 mM glucose in these experiments showed that after 10 min of a fast increase, the reducing equivalents in cultured islets increased only slowly (see Figure 2C). This may reflect the lower supply of reducing equivalents as suggested above.

To quantify mitochondrial membrane potential of cultured vs. fresh islets, the generation of a residual membrane potential by the reverse flow of protons in the F_0_ ATPase as produced by the intramitochondrial ATP hydrolysis has to be inhibited [28]. For this reason the effect of CCCP on fresh and cultured islets was characterized in the presence of oligomycin [29]. This comparison was additionally performed in the presence of the inhibitor of cytosolic protein translation, cycloheximide [30]. Based on the data, we conclude that: (i) The CCCP-induced decrease in the mitochondrial membrane potential and of the reducing equivalents and, by inference, their pre-inhibitory value is larger in cultured than in fresh islets. (ii) The decrease is augmented by the presence of cycloheximide, more so for the reducing equivalents than for the membrane potential.

The latter interpretation is consistent with a scenario where the inhibitory effect of cycloheximide on insulin secretion of fresh islets not only reflects diminished production of insulin granules or regulators of exocytosis [31], but involves changes in mitochondrial energetics [16]. Of note, the early investigations on the role of protein biosynthesis for insulin secretion see, e.g., [32,33] had all been conducted with fresh preparations. While the cycloheximide effect on the synthesis of pro-insulin and the cytosolic Ca^2+^ concentration was the same in fresh and cultured islets, the effect on the mitochondrial autofluorescence differed [16]. Correspondingly, the resulting inhibition of insulin secretion was much less efficient with cultured than with fresh islets [16].

The present measurements of the OCR confirm our recent observation that NMRI mouse islets have a higher OCR when tested immediately after isolation than when tested after 24 h of cell culture [16]. Together with the higher mitochondrial membrane potential of the cultured islets (see Figure 4), this suggests that fresh islets have a larger proton leak than cultured islets, which concurs with the larger OCR decrease in fresh islets upon the addition of rotenone and antimycin. These agents inhibit the delivery of reducing equivalents to the complex I and III, respectively, of the respiratory chain [34,35] and the resulting decrease in the OCR can be attributed to the leakiness of the inner mitochondrial membrane when the proton flow (forward or reverse) through the F_0_ ATPase is blocked by oligomycin [36].

Compared with mitochondria in other tissues, beta cell mitochondria have been shown to be of higher leakiness and as a consequence to have a lower degree of energy coupling [37]. Since these measurements were performed with cultured islets, the present data show that the leak and thus the degree of uncoupling are even higher in the freshly isolated state. This view fits to our earlier observation that the ATP/ADP ratio in cultured islets is significantly higher upon glucose stimulation [16]. The term leakiness does not imply an unregulated mechanism of action. The uncoupling protein UCP2 may seem the most probable candidate [38], see however [39], but, alternatively, the mitochondrial ATP/ADP translocase [40] or the mitochondrial permeability transition pore, of which the translocase may form part [41], can act as regulated proton-conducting pathways. Non-esterified fatty acids are usually considered as activators both of UCPs and the transition pore [41].

The leakiness of the inner mitochondrial membrane is not only a constant drain of chemically stored energy, the proton cycling can be expected to increase with the accelerated proton pumping of the respiratory chain during nutrient stimulation. This may be the underlying mechanism of the high energy-requiring step which was postulated to exist in the nutrient stimulation of insulin secretion based on OCR measurements [42,43]. The degree of leakiness which persists during cell culture may also be responsible for the marked decrease in the ATP/ADP ratio which was visible with beta cells but not alpha cells upon lowering of the ambient glucose concentration [24].

The open question for future research is whether the enhanced mitochondrial leakiness reflects non-physiological alterations sustained during the isolation procedure or whether it reflects the working condition of the islet in situ, i.e., when the islet is integrated in the tissue of the exocrine pancreas. The following considerations point to the latter possibility: The generation of ROS by the respiratory chain increases near-exponentially with the mitochondrial membrane potential [44], thus the well-known poor antioxidant defense of the beta cell [45,46] may represent an adaption to this condition. Modelling of the beta cell metabolism has suggested that physiological regulation is possible with a low ROS production as a consequence of a decreased mitochondrial membrane potential [47]. Since this is a study on mouse islets, the conclusions should not prematurely be extrapolated to human islets. However, indirect evidence that the freshly isolated state of human islets may be more robust than the cultured state comes from islet transplantation, most notably the Edmonton protocol, where islets were transplanted immediately after isolation [48,49].

## Figures and Tables

**Figure 1 biomedicines-12-01747-f001:**
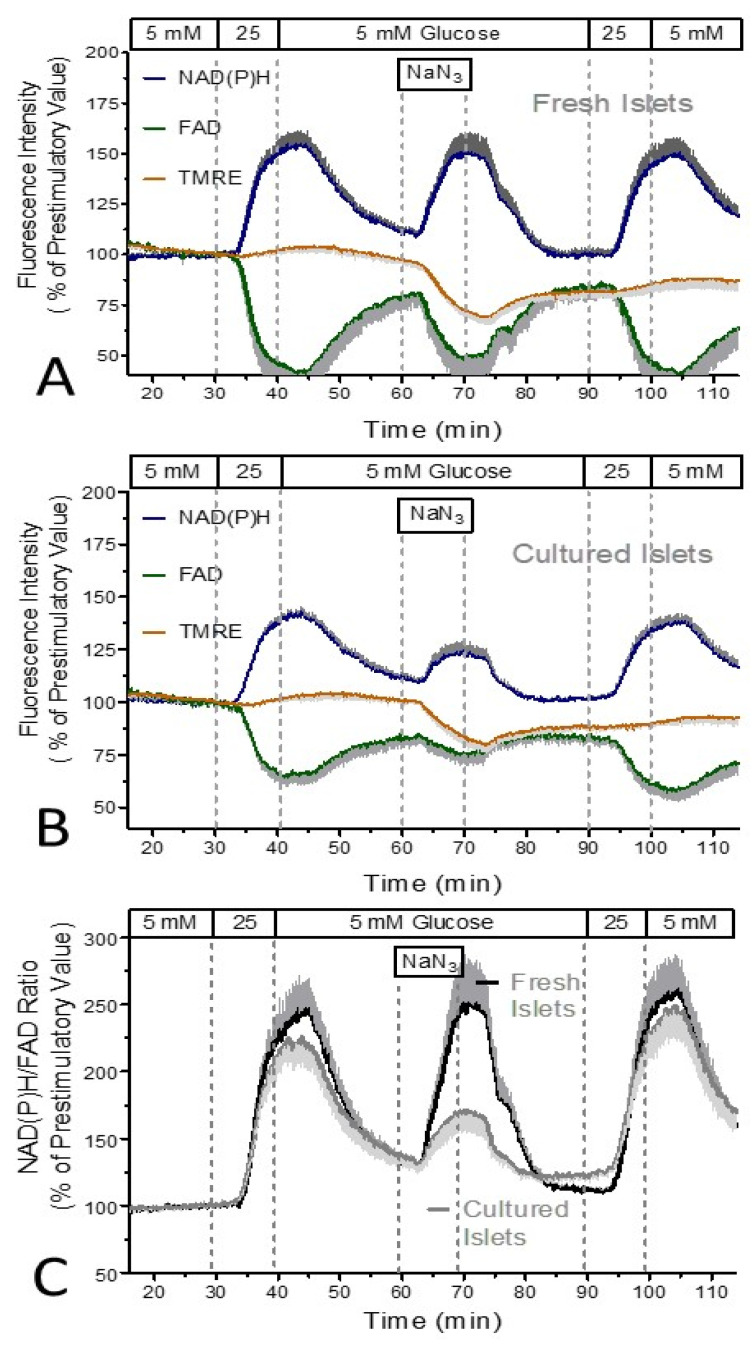
NAD(P)H- and FAD-autofluorescence and the mitochondrial membrane potential during glucose stimulation and exposure to NaN_3_. Freshly isolated islets (**A**) or cultured islets (**B**) were perifused with Krebs–Ringer medium with 5 mM glucose for 30 min, then the glucose concentration was raised to 25 mM for 10 min and was decreased back to 5 mM thereafter. After 20 min at 5 mM glucose 5 mM NaN_3_ was additionally present for 10 min. After a wash-out period of 20 min at 5 mM glucose, a second stimulation with 25 mM glucose was performed. Blue traces denote the NAD(P)H autofluorescence, the green traces the FAD autofluorescence, and the orange traces the TMRE fluorescence. For a direct comparison of the reactions of fresh and of cultured islets the NAD(P)H/FAD ratio was calculated (**C**). The black trace denotes the fresh islets, the gray trace the cultured islets. Note the much smaller increase by NaN_3_ in cultured islets and the virtually unchanged response of both types of islets to the glucose stimulation after the exposure to NaN_3_. Values are the means ± SEM of five experiments each.

**Figure 2 biomedicines-12-01747-f002:**
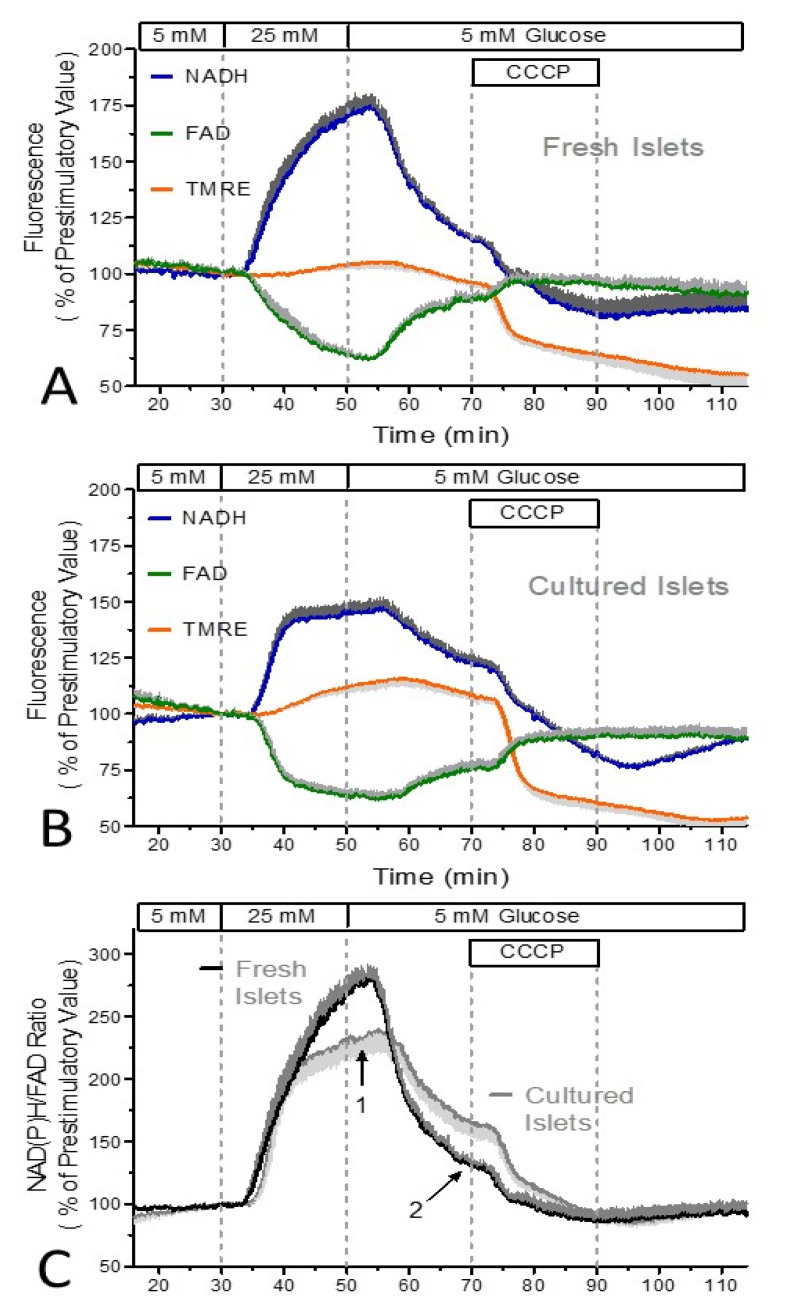
NAD(P)H- and FAD-autofluorescence and the mitochondrial membrane potential during glucose stimulation and exposure to the uncoupler, CCCP. Freshly isolated islets (**A**) or cultured islets (**B**) were perifused with Krebs–Ringer medium with 5 mM glucose for 30 min, then the glucose concentration was raised to 25 mM for 20 min and was decreased back to 5 mM thereafter. After 20 min at 5 mM glucose 10 μM CCCP was additionally present for 20 min and washed out thereafter. Blue traces denote the NAD(P)H autofluorescence, the green traces the FAD autofluorescence, and the orange traces the TMRE fluorescence. Note the significantly larger decrease in the TMRE fluorescence with cultured islets. For a direct comparison of the reactions of fresh and of cultured islets, the NAD(P)H/FAD ratio was calculated (**C**). The black trace denotes the fresh islets, the gray trace the cultured islets. Note the change from a fast to a slow increase during glucose stimulation of cultured islets (arrow 1) and the faster decrease upon wash-out of high glucose with fresh islets (arrow 2). Values are the means ± SEM of five experiments each.

**Figure 3 biomedicines-12-01747-f003:**
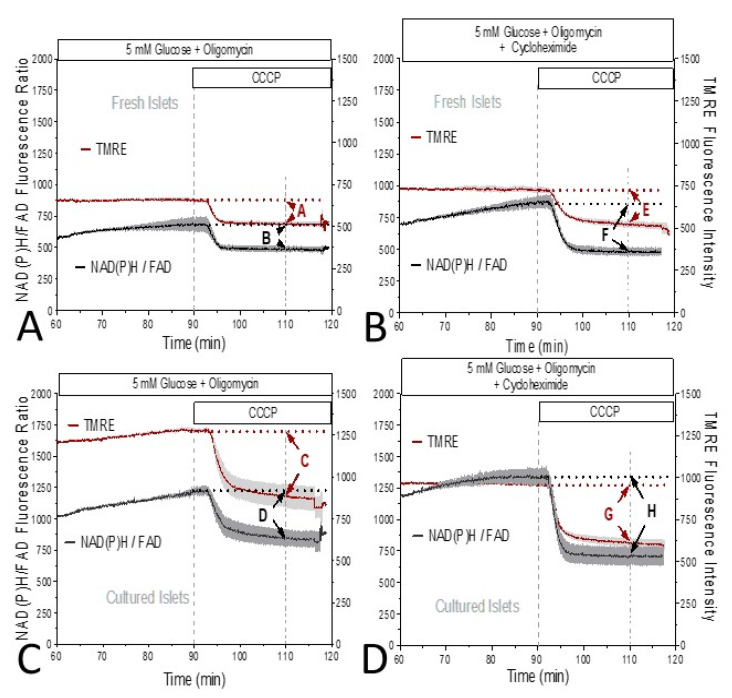
The NAD(P)H/FAD autofluorescence ratio and the mitochondrial membrane potential during exposure to oligomycin and the additional presence of the uncoupler CCCP (10 μM). Freshly isolated islets (**A**) or cultured islets (**C**) were perifused with Krebs–Ringer medium containing 5 mM glucose. From 60 min on, the ATPase inhibitor oligomycin was present, from 90 min on 10 µM CCCP was additionally present. The same protocol was used with freshly isolated islets (**B**) or cultured islets (**D**) to assess the effect of 10 μM cycloheximide, which was present throughout the entire perifusion. After normalization of the intensities to 100% at 60 min, the NAD(P)H/FAD- and the TMRE traces were multiplied with the mean intensity value at this time point to make the different datasets comparable. Values are the means ± SEM of five experiments each. For letters A–H see Figure 4.

**Figure 4 biomedicines-12-01747-f004:**
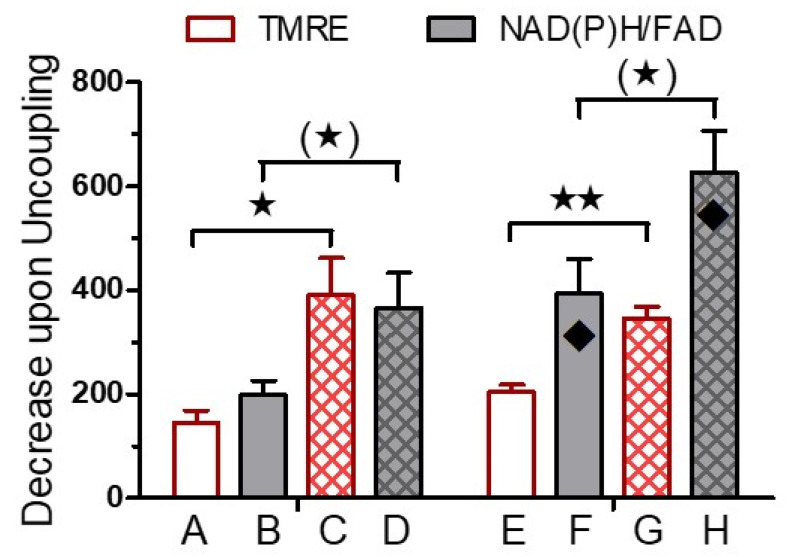
Quantitative comparison of the effect of CCCP on the NAD(P)H/FAD ratio and the mitochondrial membrane potential (TMRE) in fresh (open bars) and in cultured (cross-hatched bars) islets. The columns A to H denote the difference between the fluorescence intensity in the presence of oligomycin immediately before the addition of CCCP and the one after 20 min of exposure to CCCP as indicated by the arrows in Figure 3. Significances were calculated by *t*-test with Welch’s correction. The asterisks denote significant differences between fresh and cultured islets (^(^*^)^ *p* ˂ 0.07, * *p* ˂ 0.05, ** *p* ˂ 0.001), the rhombus symbols denote significant (*p* ˂ 0.05) differences between cycloheximide-exposed (E,F) and control (A–D) islets. Values are the means ± SEM of five experiments each.

**Figure 5 biomedicines-12-01747-f005:**
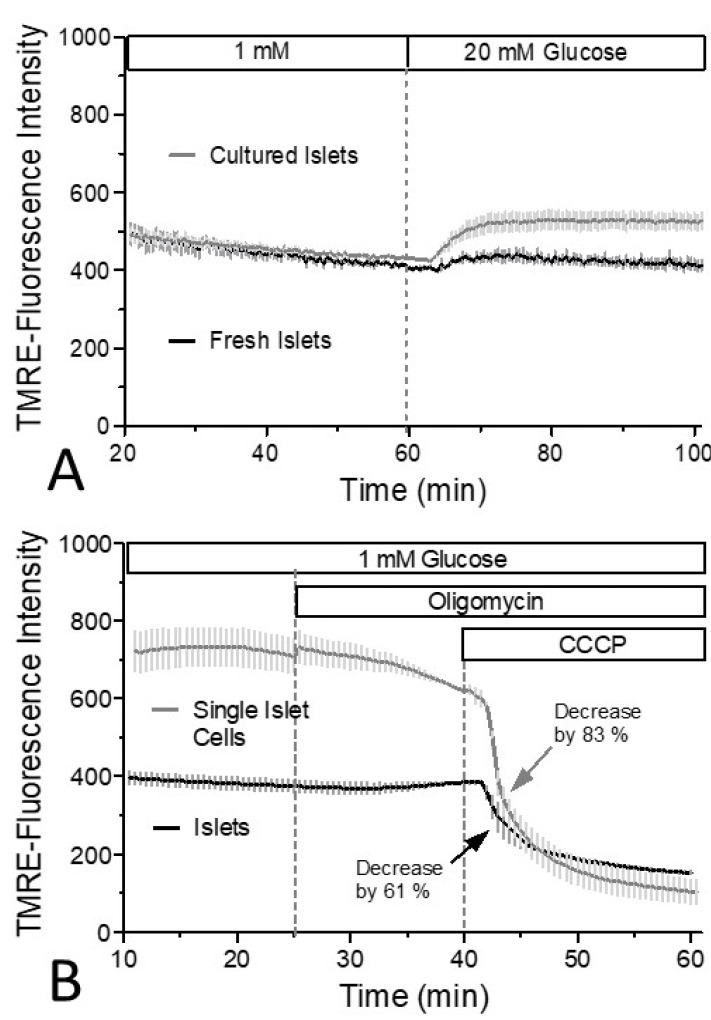
Comparison of the dynamic range of TMRE fluorescence changes. The glucose concentration in the perifusion medium of freshly isolated islets or cultured islets (**A**) was raised from a low value to a strong stimulatory value. Note the significantly larger increase in the cultured islets. Values are the means ± SEM of three experiments each. Cultured islets and single islet cells were perifused with Krebs–Ringer medium with 1 mM glucose for 25 min, then oligomycin was added and at 40 min 10 µM CCCP was added (**B**). Note the larger decrease and lower final level of the TMRE fluorescence with single cells. Values are the means ± SEM of four experiments each.

**Figure 6 biomedicines-12-01747-f006:**
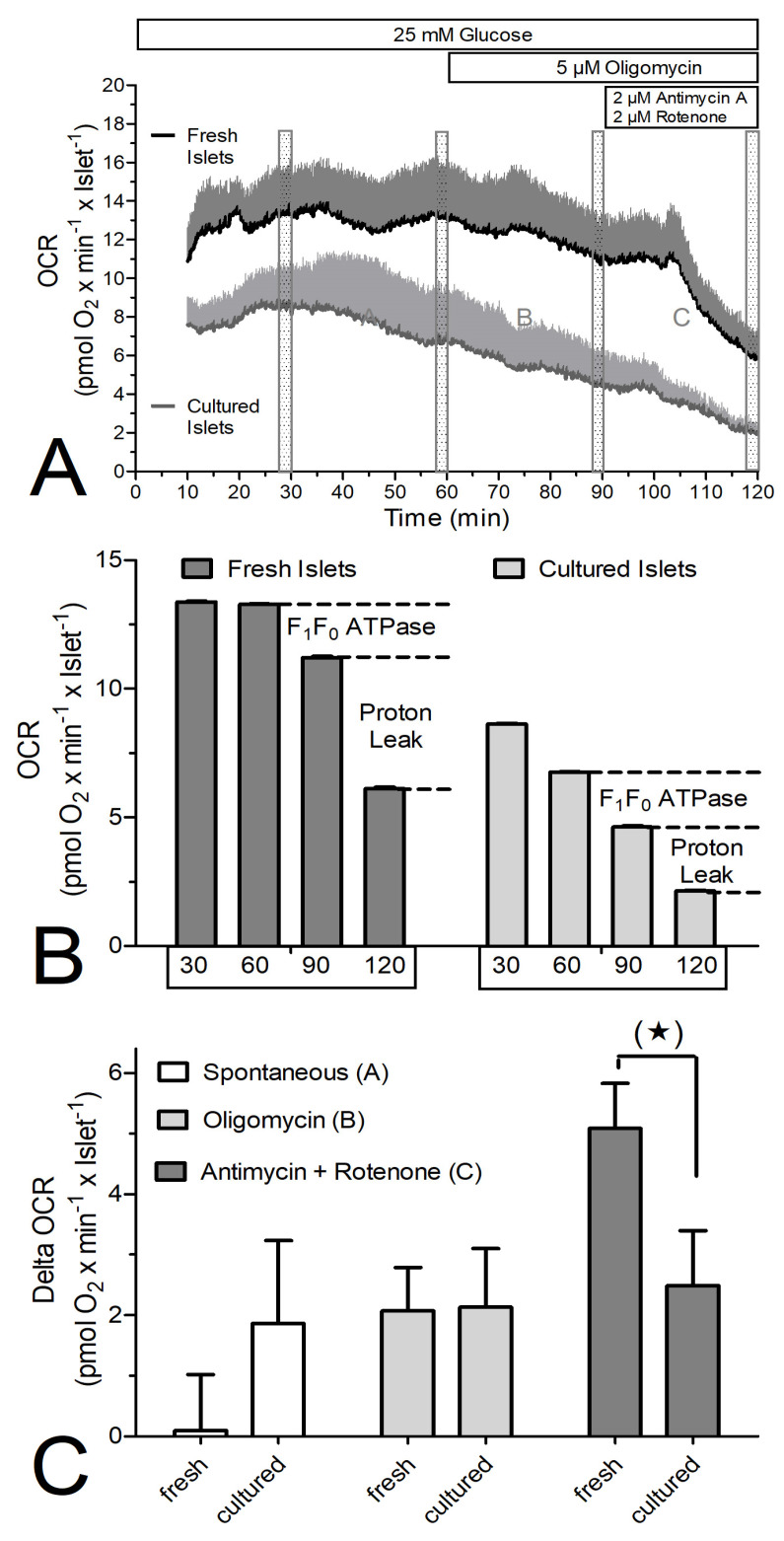
The oxygen consumption rate (OCR) of freshly isolated or cultured islets during perifusion with 25 mM glucose and inhibitors of the respiratory chain. (**A**) Freshly isolated islets (dark grey) or cultured islets (light grey) were perifused with Krebs–Ringer medium containing 25 mM glucose. From 60 min on, the ATPase inhibitor oligomycin was present, from 90 min on the inhibitors antimycin A and rotenone were additionally present. (**B**) The mean OCR during the 2 min time period prior to 30 min, 60 min, 90 min and 120 min was calculated to assess the contribution of the ATPase and of uncoupled proton flux to the total OCR. (**C**) The differences between 30 min, 60 min, 90, and 120 min (time periods A, B and C, respectively) were calculated to assess the difference between fresh and cultured islets. Asterisk denotes a marginally significant difference (^(^*^)^ *p* ˂ 0.07, paired two-sided *t*-test). Values are the means ± SEM of four experiments each.

**Figure 7 biomedicines-12-01747-f007:**
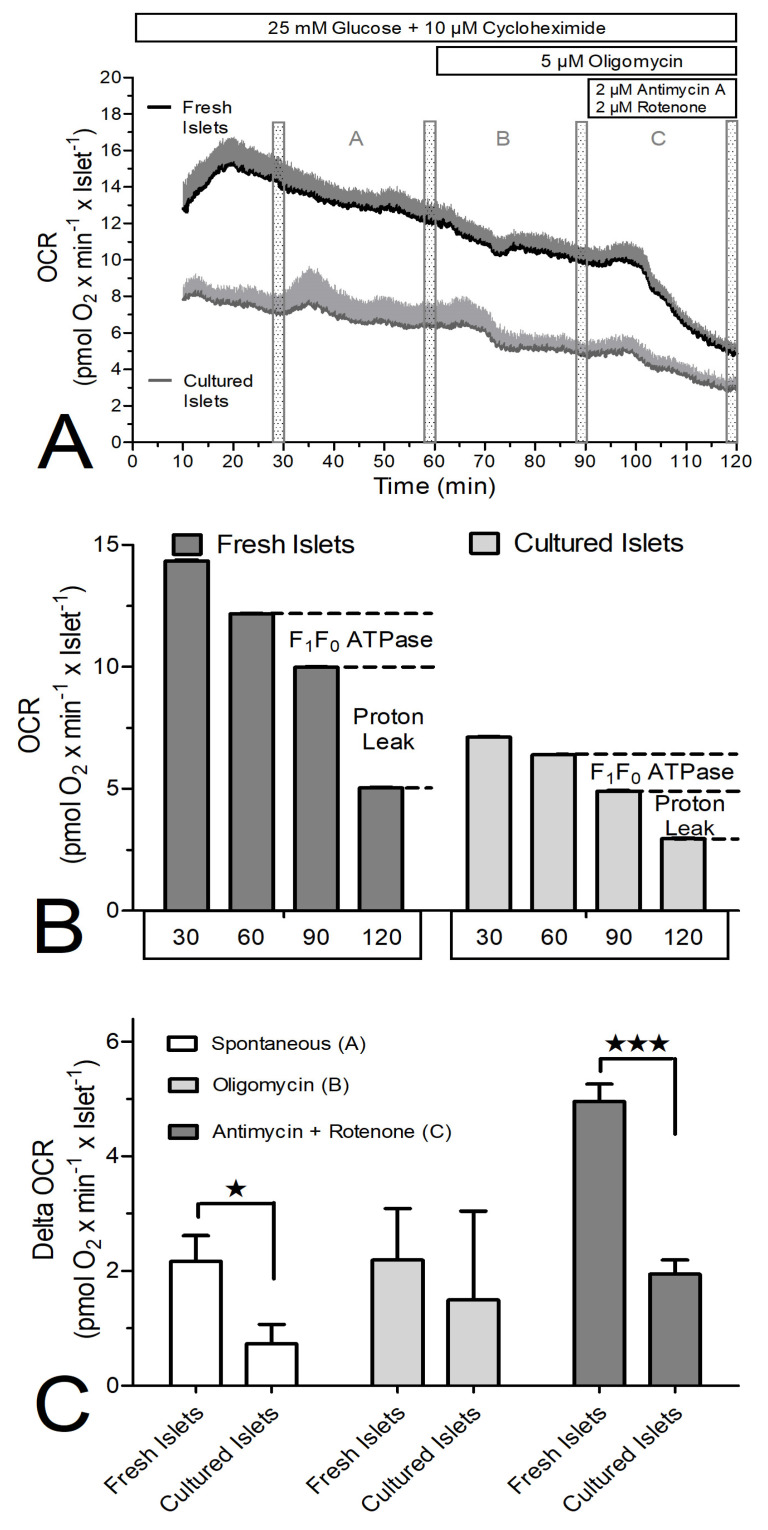
The oxygen consumption rate (OCR) of freshly isolated or cultured islets during perifusion with 25 mM glucose plus cycloheximide and inhibitors of the respiratory chain. (**A**) Freshly isolated islets (dark grey) or cultured islets (light grey) were perifused with Krebs–Ringer medium containing 25 mM glucose and 10 μM cycloheximide. From 60 min on, the ATPase inhibitor oligomycin was present, from 90 min on the inhibitors antimycin A and rotenone were additionally present. (**B**) The mean OCR during the 2 min time period prior to 30 min, 60 min, 90 min and 120 min was calculated to assess the contribution of the ATPase and of uncoupled proton flux to the total OCR. (**C**) The differences between 30 min, 60 min, 90 min and 120 min (time periods A, B and C, respectively) were calculated to assess the difference between fresh and cultured islets. The asterisks denote significant differences (* *p* ˂ 0.05, *** *p* ˂ 0.001, paired two-sided *t*-test). Values are the means ± SEM of five experiments each.

## Data Availability

The data sets generated during and/or analyzed during the current study are not publicly available but are available from the corresponding authors on reasonable request.

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
