# Peer review of "The Proton Leak of the Inner Mitochondrial Membrane Is Enlarged in Freshly Isolated Pancreatic Islets"

_biomedicines, 2024, doi:10.3390/biomedicines12081747_

Round 1

Reviewer 1 Report

Comments and Suggestions for Authors

The study by Alshafei et al. investigated the mitochondria function in islets that were with or without in vitro culture. The freshly isolated islets were used within an hour after isolation, and a 22-hour culture was used to establish the ‘cultured islets’ preparation. The authors tested the mitochondrial membrane potential of the different preparations and found a lower potential in fresh islets and they concluded that is due to a higher proton leak. This is an interesting study that aims to address an important technical aspect of islet research – most of the studies utilise cultured islets and previous research did notice the functional impact of culture (including the works by the authors). The manuscript is well-written and presented in a logical order. I have a few comments as follows.

Major

1.    Whereas the evidence presented shows the changes in mitochondria function, it is unclear how the culture reduced proton leak (if as authors suggested a higher leakage is more physiological). Authors should discuss possible underlying mechanisms.

2.    The 45-minute window is very narrow and the isolation process is very stressful for islets. It is unclear whether the ‘fresh’ islets were exposed to the culture medium at all. If not, Kreb’s solution, though physiological, is only saline and does not contain some essential nutrients and growth factors. Would this be underlying the changes the authors observed?

3.    Another question is the time course of the functional change. The authors used a single time point (22 hours). Many laboratories use 2- 4 hours post-isolation islets. It may be of interest to establish a time course of the changes induced by culture.

4.    It is unclear what are the implications for human islets, both in terms of research and clinical application. Human islets are typically used after culture. Would the authors’ study indicate that a much shorter culture time is required for better clinical and research outcomes?

Minor

1.    There are a few typos in the text.

2.    It is normally accepted a p value <0.05 marks significance. However, there are several places in the manuscript, the significance level was defined as p<0.07. It requires justification/explanation.

Author Response

The study by Alshafei et al. investigated the mitochondria function in islets that were with or without in vitro culture. The freshly isolated islets were used within an hour after isolation, and a 22-hour culture was used to establish the ‘cultured islets’ preparation. The authors tested the mitochondrial membrane potential of the different preparations and found a lower potential in fresh islets and they concluded that is due to a higher proton leak. This is an interesting study that aims to address an important technical aspect of islet research – most of the studies utilise cultured islets and previous research did notice the functional impact of culture (including the works by the authors). The manuscript is well-written and presented in a logical order. I have a few comments as follows.

We thank the reviewer for his/her careful reading and insightful comments

Major

  1.   Whereas the evidence presented shows the changes in mitochondria function, it is unclear how the culture reduced proton leak (if as authors suggested a higher leakage is more physiological). Authors should discuss possible underlying mechanisms.

Actually, two structures which may serve as correlates of the proton leak are mentioned, namely the uncoupling proteins, in particular UCP2 and a structure which is rarely mentioned outside mitochondrial research, the permeability transition pore. We have now added a sentence to mention the most likely regulators of either structure, the non-esterified fatty acids.

  1.   The 45-minute window is very narrow and the isolation process is very stressful for islets. It is unclear whether the ‘fresh’ islets were exposed to the culture medium at all. If not, Kreb’s solution, though physiological, is only saline and does not contain some essential nutrients and growth factors. Would this be underlying the changes the authors observed?

Frankly, the question whether factors which influence the mitochondrial degree of coupling are lost by perifusion with Krebs-Ringer solution or are taken up from the cell culture medium is still wide open. However, there may be additional factors which may influence the islet function: the islet situated in a conventional Petri dish filled with RPMI plus FCS is within a microenvironment with reduced oxygen tension and increased concentration of released metabolites. This why constant perifusion by islet on chip systems is becoming more popular.

  1.   Another question is the time course of the functional change. The authors used a single time point (22 hours). Many laboratories use 2- 4 hours post-isolation islets. It may be of interest to establish a time course of the changes induced by culture.

This is the task of a master student who currently works in our lab. Using the experimental approach as shown in figure 5, she finds that the TMRE fluorescence increase following glucose stimulation is virtually the same after 24 h and 48 h but is getting less after 72 h. Similarly, the decrease caused by CCCP in the presence of oligomycin is diminished after 72 h. Years ago Henquin´s groups has shown a degradation of the typical Ca2+ response pattern upon prolonged islet culture.

  1.   It is unclear what are the implications for human islets, both in terms of research and clinical application. Human islets are typically used after culture. Would the authors’ study indicate that a much shorter culture time is required for better clinical and research outcomes?

There is a clinical observation that a short time span between islet isolation and transplantation is a favorable condition for islet survival: this is the Edmonton transplatation protocol. While there were additional features which may be responsible for the increased success rate, such as immunosuppression without glucocorticoids, it is also relevant that the islets were transplanted within hours after the isolation. This view was supported by independent investigations. This fact is now mentioned in the discussion section and two references have been added.

Minor

  1.   There are a few typos in the text.

The text has been checked and several typos have been corrected.

  1.   It is normally accepted a p value <0.05 marks significance. However, there are several places in the manuscript, the significance level was defined as p<0.07. It requires justification/explanation.

Actually, it is the Prism software which has drawn my attention to the topic of marginally significant p values. The p value is a continuous variable, considering p values smaller than 0.05 as significant and those larger than 0.05 as non-significant is simply a convention. Mentioning p values which are marginally significant may be useful to emphasize a response pattern. In the present ms this e.g. concerns the changes caused by antimycin and rotenone in the OCR experiments (figures 6 and 7). While the reduction is strongly significant with cultured islets it is clearly less so with freshly isolated islets, but it would be a virtual misrepresentation to just describe the reduction as non-significant which is often considered as equivalent to non-existent.

Reviewer 2 Report

Comments and Suggestions for Authors

The study describes a detailed experimental analysis of the coupling of the electron transport chain in isolated mouse pancreatic islets, with a view to investigating the metabolic amplification of insulin secretion.  The study would provide mechanistic insights into this important second phase of insulin release.  Despite more than 60 years of research in this field there are still some fundamental questions to be resolved, perhaps reflecting the difficulties of establishing a realistic and accessible experimental model of stimulation-secretion coupling of insulin release.  The model used has the attractive feature of being a primary culture of an intact pancreatic islets from an established strain of mice.  The islets would be expected to contain beta cells predominantly and so inferences about data are interpreted accordingly.

The background and discussion of the study are described very clearly, supported by strong references.  Some are of the corresponding author's previous work but these are mainly directly related to the study.

The methods are described clearly.  The results have been carefully arranged and summarise a lot of data very clearly.  The only slight deficiency was the labelling in Figure 4 which could show which data are from freshly isolated islets and which are from cultures islets, e.g by changing the shading of the bars.  The comprehensive figure legends are particularly helpful.

It was of interest to note the effect of cycloheximide on the spontaneous changes in OCR (Fig.6c compared with Fig. 7c)  - a comment of the potential reason for this, and for the use of cycloheximide generally might be helpful.

The overall conclusion of increased leakiness of protons across the inner  mitochondrial membrane in fresh versus cultured islets could have important implications for other experimental models, especially commonly used beta cell lines.  

Mild uncoupling of proton gradients may occur for some proposed new, phyto-, treatments, albeit targeting other tissues, but effects on beta cells need to be monitored too.  The study emphasises the importance of investigating mitochondrial mechanisms thoroughly and is a worthwhile contribution to the field.  

Author Response

This paper aims to investigate the cause of the difference in secretion between freshly isolated islets and islets cultured for one day. The results indicate that this difference is not linked to cell viability but to the signaling of insulin secretion. This study is interesting but requires modifications:

We thank the reviewer for his/her critical reading and have dealt with the points as follows

  1. The abstract is generally written. It is very important to include the results as values with statistical significance, as well as the effect of inhibitors of oxidative phosphorylation on all measured indices.

Please consider that most measurements reported fluorescence values and the logic of the interpretation is based on relative changes. Such type of observations is best described in general terms. However, the measurement of the oxygen consumption gives values which permit a meaningful comparison with values obtained by other labs. This is why we have now included a numerical comparison of the OCRs of fresh and of cultured islets, which are significantly different.

  1. Line 132: Please delete the references in the results section.

We agree with the reviewer that the results section should not be stuffed with references, as it detracts the reader from critically evaluating the data. However, in the present ms there is only one reference in the results section and it is placed in the introductory paragraph to underline the justification of why we used sodium azide as the first mitochondrial inhibitor. In our feeling it would have been misplaced hade we mentioned it only later in the discussion section.

  1. The results must be presented as values with percentages or statistical analysis.

The normalized fluorescence intensity values in figures 1 and 2 can be considered to be equivalent to percentage values. The statistical evaluation of the fluorescence values of figure 3 is shown in figure 4 and figures 6c and 7c contain the statistical evaluations of the original registrations as shown in figures 6a and 7a, respective. When a difference between test and control is outright obvious, like the different effects of sodium azide in figure 1c, a statistical evaluation does not yield additional information for the reader.

  1.  the presentation of results with titles and subtitles is clearer for readers.

We have considered subdividing the results section before submitting the original version and have rechecked now, but have come again to the conclusion that except for figures 3 and 4 and figures 6 and 7, a separate chapter would be needed for each single figure. In our opinion this would not increase the readability. 

  1.    In the conclusion, the limitations of this study added

 Of course, this is a study on mouse islets and its results should not be prematurely extrapolated to be equally valid for human islets. However, there is indirect evidence that freshly isolated human islets are more tolerant to the stress of transplantation, which fits to the logic of the present investigation (see also answers to reviewer 1). These aspects are now mentioned at the end of the discussion section.

  1. The similitude rate is 35%; must be less than 20%

This study is a continuation of the investigation by Alshafei et al (2023) and uses the same methodology. For this reason the materials and methods section is nearly identical and initial parts of the introduction are quite similar. To comply with the reviewer´s suggestion we have rephrased the wording in the materials and methods section. 

Reviewer 3 Report

Comments and Suggestions for Authors

Manuscript ID : biomedicines-3067894

This paper aims to investigate the cause of the difference in secretion between freshly isolated islets and islets cultured for one day. The results indicate that this difference is not linked to cell viability but to the signaling of insulin secretion. This study is interesting but requires modifications:

 1.    The abstract is generally written. It is very important to include the results as values with statistical significance, as well as the effect of inhibitors of oxidative phosphorylation on all measured indices.

2.    Line 132: Please delete the references in the results section.

3.    The results must be presented as values with percentages or statistical analysis.

4.  the presentation of results with titles and subtitles is clearer for readers.

5.    In the conclusion, the limitations of this study added

6. The similitude rate is 35%; must be less than 20%

Author Response

The study describes a detailed experimental analysis of the coupling of the electron transport chain in isolated mouse pancreatic islets, with a view to investigating the metabolic amplification of insulin secretion.  The study would provide mechanistic insights into this important second phase of insulin release.  Despite more than 60 years of research in this field there are still some fundamental questions to be resolved, perhaps reflecting the difficulties of establishing a realistic and accessible experimental model of stimulation-secretion coupling of insulin release.  The model used has the attractive feature of being a primary culture of an intact pancreatic islets from an established strain of mice.  The islets would be expected to contain beta cells predominantly and so inferences about data are interpreted accordingly.

We thank the reviewer for his/her careful reading and the favorable assessment of our work.
Our response to the specific points is as follows:

The background and discussion of the study are described very clearly, supported by strong references.  Some are of the corresponding author's previous work but these are mainly directly related to the study.

The methods are described clearly.  The results have been carefully arranged and summarise a lot of data very clearly.  The only slight deficiency was the labelling in Figure 4 which could show which data are from freshly isolated islets and which are from cultures islets, e.g by changing the shading of the bars.  The comprehensive figure legends are particularly helpful.

We have followed the reviewer´s advice and have cross-hatched the bars representing the values obtained with cultured islets.

It was of interest to note the effect of cycloheximide on the spontaneous changes in OCR (Fig.6c compared with Fig. 7c)  - a comment of the potential reason for this, and for the use of cycloheximide generally might be helpful.

As inhibitor of the cytosolic transplantation machinery cycloheximide has been used to inhibit the glucose-dependent synthesis of insulin. However, in our recent publication we have come to the conclusion that cycloheximide does not only decrease the amount of newly formed proinsulin but at the same time also affects mitochondrial function. The likely explanation for this unexpected finding is that the vast majority of mitochondrial proteins is produced by cytosolic translation and thereafter imported by the mitochondria. We have inserted two sentences on this topic in the discussion.

The overall conclusion of increased leakiness of protons across the inner  mitochondrial membrane in fresh versus cultured islets could have important implications for other experimental models, especially commonly used beta cell lines.  

Mild uncoupling of proton gradients may occur for some proposed new, phyto-, treatments, albeit targeting other tissues, but effects on beta cells need to be monitored too.  The study emphasises the importance of investigating mitochondrial mechanisms thoroughly and is a worthwhile contribution to the field.  

Reviewer 4 Report

Comments and Suggestions for Authors

This study presents interesting findings on the metabolic differences between freshly isolated and cultured islets, with potential implications for understanding insulin secretion and mitochondrial function. The authors observed that the mitochondrial membrane potential is lower and the exchange of mitochondrial reducing equivalents is faster in freshly isolated islets compared to cultured islets. Additionally, a significantly larger proton leak exists in fresh islets, which accounts for their higher oxygen consumption rate. The authors conclude that these metabolic differences may explain the variations in insulin secretion observed between the two types of islets.

The authors need to clarify how the lower mitochondrial membrane potential and faster exchange of reducing equivalents in fresh islets contribute to the observed differences in insulin secretion.

Discuss the potential physiological relevance of the larger proton leak in fresh islets and how it might relate to the in vivo environment within the pancreas.

Since this study involves islets isolated from the pancreas of female NMRI mice, the details of ethical approval need to be mentioned.

Comments on the Quality of English Language

English looks good

Author Response

This study presents interesting findings on the metabolic differences between freshly isolated and cultured islets, with potential implications for understanding insulin secretion and mitochondrial function. The authors observed that the mitochondrial membrane potential is lower and the exchange of mitochondrial reducing equivalents is faster in freshly isolated islets compared to cultured islets. Additionally, a significantly larger proton leak exists in fresh islets, which accounts for their higher oxygen consumption rate. The authors conclude that these metabolic differences may explain the variations in insulin secretion observed between the two types of islets.

We thank the reviewer for his/her careful reading and the constructive remarks

The authors need to clarify how the lower mitochondrial membrane potential and faster exchange of reducing equivalents in fresh islets contribute to the observed differences in insulin secretion.

From the chemiosmotic theory of oxidative phosphorylation it can be expected that a “wasteful” return of protons into the mitochondrial matrix without passing the proton channel of the F0 ATPase should diminish nutrient-induced increases of the ATP/ADP ratio. The significantly smaller increase of the ATP/ADP ratio in fresh islets has been described in the preceding publication Alshafei et al. 2023 (ref 16) as is mentioned in the discussion section. Consequently, the triggering signal and the size of the first phase will be more prominent in cultured islets. The persistent prominence of the first phase in cultured islets has been observed by Morsi et al. 2020 (ref 15).

Discuss the potential physiological relevance of the larger proton leak in fresh islets and how it might relate to the in vivo environment within the pancreas.

The exocrine pancreas can be quite fatty and the open question is whether this creates a microenvironment for the islets which favors the increased proton leak. However these are quite speculative deliberations. To illustrate the physiological relevance we have added a sentence at the very end of the discussion about the higher robustness of freshly isolated islets as evidenced by data from the islet transplantation.

Since this study involves islets isolated from the pancreas of female NMRI mice, the details of ethical approval need to be mentioned.

In Germany the permission to perform animal experimentation is not given by a university ethics committee but by an independent external institution (which is more critical). However, sacrifice of rodents to perform ex vivo experiments is not considered animal experimentation, thus the external institution (the LAVES for the state of Lower Saxony) only controls the animal facility (are the mice kept in an enriched environment, is the personnel qualified, is the book-keeping correct) and requires annual data of how many mice were kept and how many of them were actually sacrificed. Thus no project-specific registration number exists. The description in the manuscript is accurately correct.